# Assessing the Health Risk and the Metal Content of Thirty-Four Plant Essential Oils Using the ICP-MS Technique

**DOI:** 10.3390/nu14122363

**Published:** 2022-06-07

**Authors:** Andreea Maria Iordache, Constantin Nechita, Cezara Voica, Carmen Roba, Oana Romina Botoran, Roxana Elena Ionete

**Affiliations:** 1National Research and Development Institute for Cryogenics and Isotopic Technologies, ICSI, 4 Uzinei Street, 240050 Ramnicu Valcea, Romania; iordache_andreeamaria@yahoo.com (A.M.I.); oana.dinca@icsi.ro (O.R.B.); roxana.ionete@icsi.ro (R.E.I.); 2National Research and Development Institute for Forestry “Marin Dracea” Calea Bucovinei, 73 Bis, 725100 Campulung Moldovenesc, Romania; 3National Institute for Research and Development of Isotopic and Molecular Technologies, 67-103 Donath, 400293 Cluj-Napoca, Romania; 4Biomolecular Physics Department, Faculty of Physics, Kogalniceanu 1, Babes–Bolyai University, 400084 Cluj-Napoca, Romania; carmen.roba@ubbcluj.ro

**Keywords:** essential oils, heavy metals, ICP-MS technique, dietary assessment risk, macronutrients, micronutrients, Romanian markets

## Abstract

Natural ecosystems are polluted with various contaminants, and among these heavy metals raise concerns due to their side effects on both environment and human health. An investigation was conducted on essential oil samples, comparing similar products between seven producers, and the results indicated a wide variation of metal content. The recommended limits imposed by European Union regulations for medicinal plants are exceeded only in *Mentha* × *pipperita* (Adams, 0.61 mg/kg). Except for *Thymus vulgaris*, the multivariate analysis showed a strong correlation between toxic and microelements (*p* < 0.001). We verified plant species–specific bioaccumulation patterns with non-metric multidimensional scaling analysis. The model showed that Adams, Doterra, Hypericum, and Steaua Divina essential oils originated from plants containing high micro and macroelement (Cu, Mn, Mg, Na) levels. We noted that the cancer risk values for Ni were the highest (2.02 × 10^−9^–7.89 × 10^−7^). Based on the target hazard quotient, three groups of elements were associated with a possible risk to human health, including As, Hg, and Cd in the first group, Cr, Mn, Ni, and Co in the second, and Zn and Al in the third. Additionally, the challenge of coupling inter-element relationships through a network plot analysis shows a considerable probability of associating toxic metals with micronutrients, which can address cumulative risks for human consumers.

## 1. Introduction

Plant essential oils (EOs) are commercially used in medicine, flavoring agents in foods, and flavor and insecticides in spices, and they are intensively studied for their antioxidant, antibacterial, and anti-inflammatory activity [1,2]. Plant essential oils have various proprieties such as contact and fumigation toxicity, growth regulators, repellent activity, acaricides, and insecticides [3,4,5,6]. The EOs are a complex mixture of low molecular weight (<500 Daltons), representing volatile organic compounds resulting as secondary metabolites encapsulated in multiple matrices (oils, resin, glands, or trichomes extracts from plants) [7]. The different active functional groups in plant essential oil structures are alcohol, ketone, aldehyde and lactone from terpenoids and phenylpropanoids, corresponding to stabilization and reduction [8]. The highest number of studies indicated no more than two or three essential components bioactivities from one EOs [9]. The combined actions and synergic effects of various molecules derived from multiple components are also frequently discussed in the literature [10]. Recent studies recommended the EOs for SARS-CoV-2 infection treatment due to the immunomodulatory, bronchodilatory, or antimicrobial activities during various stages of the disease [11]. Combined with the lipophilic action of EOs, essential oils can penetrate the viral membranes, and it acts positively in strengthening the human body against infections [12]. The taxonomy of plants producing essential oils belongs to multiple genera, counting around 60 families; the most known are *Alliaceae*, *Apiaceae*, *Asteraceae*, *Geraniaceae*, *Lamiaceae*, *Lauraceae*, *Liliaceae*, *Myrtaceae*, *Poaceae*, *Pinaceae*, *Rutaceae* or *Rosaceae* [13,14,15]. However, only a reduced percentage of species are used in aromatherapy, flavor, cosmetic, animal feed, and pharmaceutical industries [16].

Considering the role of EOs in alternative medicine, it is mandatory to evaluate their quality and safety. The complex beneficial role of essential oils can be underestimated due to adulteration (a process driven by independent and isolated economic interest) [17] and contamination with various pollutants (a complex action that affects all living organisms, especially plants) [18]. Environmental stress (such as climate and pollution) increases the production of the internal plant secondary metabolites as oils [19]; still, the process is poorly understood [20]. The metal origins in essential oils are associated with plant contamination, which varies with plant species, climate regime, soil composition, plant age, harvesting period, or geographical origin [21]. Therefore, the sources of contaminants are anthropogenic and less orogenic, and the pathways for chemical compounds in plant organs are mainly from the soil through the root system [22]. The water and air pollution drivers are less discussed. The transfer of metals from leaves, barks, roots, seeds, flowers, and fruits to essential oils depends on extraction technology [23]. The literature mentioned that Fe, Cu, Ag, and Zn are notoriously associated with antimicrobial, antioxidant, and other positive biological activity [24], but As, Cd, Pb, and Hg cause toxic effects at relatively low levels [25,26].

Overall, the objectives of the present study were as follows: (i) determining the concentration of seven microelements: aluminum, chromium, manganese, cobalt, nickel, copper, and zinc (Al, Cr, Mn, Co, Ni, Cu and Zn), four toxic elements: arsenic, cadmium, mercury and lead (As, Cd, Hg and Pb), and two macroelements: manganese and sodium (Mg and Na) in 34 plant essential oils with various world origins accessible in the Romanian market; (ii) to analyze and to compare the quantified concentrations with recommended safe limits; (iii) to evaluate the health risk assessment using chronic daily intake (CDI) and total hazard quotient (THQ); (iv) understanding the association between microelements, macroelements, and toxic metals in essential oils, and (v) to test hypotheses related to potential origins of natural absorption in plants from the environment or if the essential oil enrichment was related to extraction process.

## 2. Materials and Methods

### 2.1. Chemical Materials and Preparation

The chemical content quality of five plant essential oils with seven producers is accessible in the Romanian market. They were investigated, considering that each distributor provides quality certificates to ensure quality products (Table 1). Samples from five different bottles were combined for each producer/essential oil to minimize the effect of accidental extreme values. The quantification of trace elements in plant oil is challenging due to the matrix’s low levels and organic nature [27]. Numerous preparation methods such as microwave-assisted digestion [28], dry ashing [29], microwave-induced combustion [30], and extraction [31] have been used for oil-matrix sample analysis. We used ICP-MS microwave-assisted digestion to assess element levels [32]. A closed iPrep vessels speed iwaveJ system MARS6 CEM One Touch was used for this procedure. Approximately 0.5 g aliquot of each sample was weighed, followed by digestion in a mixture of (10 mL HNO_3_ % + 2 mL H_2_O_2_ %) at high pressure, temperature, and the combination of the microwave. Vessels were closed, placed on the rotor, and the microwave temperature was increased first up to 160 °C for 15 min and second to 210 °C; this final temperature was maintained for 15 min. The vessels were cooled and carefully opened at the end of the digestion process. Each digest was transferred quantitatively with ultrapure water to a 50 mL volumetric flask. These solutions were quantitatively analyzed by ICP-MS, using external standard calibration and quality control standards, respectively; for each sample, we performed ten replicates. Calibration curves were obtained for aqueous reference solutions for all the analytes, and this linearity was considered acceptable (R > 0.999). Several parameters were evaluated to validate the analytical methods for determining the studied elements in essential oil samples: (i) the limits of detection (LOD) and of quantification (LOQ) were experimentally calculated as ten and three times of standard deviation (SD) of blank determination; (ii) the accuracy (as recovery) and precision (as relative standard deviation [RSD]) of the procedure were determined by analyzing a certified reference material. Essential oils are extracts from medicinal herbs; thus, it was appropriate to choose two plant-specific certificate reference materials (CRM NCS ZC85006 tomato-trace elements and CRM IAEA-359 cabbage trace elements, China National Analysis Center for Iron and Steel, Beijing, China). The results showed that the data obtained by ICP-MS were statistically assured for a 95% confidence interval. Recoveries of elements in the CRM were between 90 and 99%, and the relative standard deviations were less than 10%.

### 2.2. Chemical and Reagents

All reagents were of analytical grade. Nitric acid (HNO_3_, ultrapure grade, 60% *w*/*w*, Merck, Darmstadt, Germany) and supra-pure hydrogen peroxide (H_2_O_2_, supra-pure, 30% *w*/*w*, Supelco, Sigma Aldrich, Darmstadt, Germany) were used for the sample digestion. Ultra-pure deionized water (18 MΩcm^−1^) from a Milli-Q analytical reagent-grade water purification system (Millipore, Darmstadt, Germany) was used for preparing all reagents and standard solutions. High-purity ICP Multielement Standard Solution XXI CertiPUR obtained from Merck (Darmstadt, Germany) was used for the calibration curve in the quantitative analysis, 10 mg/L Al, As, Cd, Co, Cr, Cu, Mg, Mn, Na, Ni, Pb, and Zn. For Hg, Mercury Standard for ICP Perkin Elmer 10 mg/L Hg in nitric acid (Atomic Spectroscopy Standard) was used. All plastic wares were cleaned by soaking for 24 h in 10% HNO_3_, followed by washing with ultrapure water. Plastic containers, pipettes, and reagents in contact with the samples or standards were randomly checked for contamination. The gaseous argon used to form the plasma in the ICP-MS was of purity 6.0.

### 2.3. Analytical Performance of the Method

The elements profile of selected essential oils was analyzed using an inductively coupled plasma quadrupole mass spectrometer ELAN DRC-e, Perkin Elmer. The operational conditions were optimized using a tuning solution (Elan 6100 Set-up/Stab/Masscal Solution 10 μg L^−1^ Ba, Cd, Ce, Cu, In, Mg, Pb, Rh, U, from Perkin Elmer, Waltham, MA, USA). Dynamic reaction cell technology (DRC) minimized interferences by placing a pressurized closed-cell between the ion lens and the quadrupole mass analyzer (QMS) in the ICP-MS. Cu and Cr were determined using the DRC mode with methane at 0.8 L min^−1^ in the reaction chamber and a rejection parameter value of 0.7. For each sample analysis, three replicates were measured to assure the quality control of measurements. We set specific parameters for controlling the performance of the ICP-MS analysis for nebulizer gas flow rate (0.92 L/min), auxiliary gas flow (1.20 L/min), plasma gas flow (15 L/min), lens voltage (12.5 V), RF power (1250 W), CeO/Ce ratio (0.03), and Ba^++^/Ba^+^ (0.014).

### 2.4. Characterization of Possible Health Risks Associated with Essential Oil Ingestion

For the products with internal usage recommendations from the producer, we evaluated the potential health risk associated with the presence of trace, micro, and macroelements elements in the essential oils. The assessment of possible health risks was performed according to the method proposed by the US-EPA [33,34]. The possible non-carcinogenic risk was estimated based on the target hazard quotients (THQs) (Equation (1)). Evaluating the cumulative potential threat caused by exposure to a mixture of trace elements does not appear to be modifying the subject of the chronic hazard index (HI). When HI < 1, it is considered a safe level, while HI ≥ 1 is a level of concern [33].
THQ = CDI/R_f_D,
CDI = (C × IR × EF × ED)/(BW × AT)(1)

The daily ingestion rate used to calculate the chronic daily intake (CDI) in µg/kg bw/day and the THQ were mentioned by the manufacturer in the product leaflet. It should be noted that in the calculation formula for CDI, the value of 350 days/year proposed by the US-EPA for the frequency of exposure (EF) has been replaced by the value of 182 days/year, as manufacturers generally recommend that the oils be administered for 7–14 days. Then, the treatment can be repeated after a break of two weeks. The R_f_Dis, the chronic reference dose of the heavy metal, in our case, are as follows 1000 (Al), 3 (Cr^IV^), 140 (Mn), 43 (Co), 20 (Ni), 40 (Cu), 300 (Zn), 0.304 (As), 1 (Cd), 0.571 (Hg), 3.57 (Pb) µg/kg bw/day. The present study considered an average body weight (BW) of 60 kg in two exposure duration testing models, respectively, 30 years were associated with non-carcinogenic risk and 70 years with carcinogenic risk. The exposure frequency (EF) was reduced to 182 days/year, and the average time lifespan (AT) was equivalent for 365 *×* 30 = 10,950 days (non-carcinogenic), respectively, 365 *×* 70 = 25,550 days (carcinogenic). 

The carcinogenic risk for lifetime cancer risk (CR) indicated the probability of individual lifetime health risk from carcinogens. It was estimated using Equation (2), where the CSF represents the slope factor of hazardous metals with the following values 0.84 (Ni), 1.5 (As), 0.38 (Cd), 8.5 × 10^−6^ (Hg), 0.084 Pb) mg/kg/day.
CR = CDI × CSF(2)

According to US-EPA recommendations, the permissible level of CR is 1 × 10^−6^ (1 in 1,000,000) to 1 × 10^−4^ (1 in 10,000). We evaluated the accumulative cancer risk caused by exposure to multiple carcinogenic elements present in essential oils, representing the sum of CR. The indices were calculated by summing the values for Ni and CR, As and CR and Cd, CR and Hg, and CR and Pb, respectively [33].

### 2.5. Statistical Analysis

Descriptive data analysis, including mean, minimum, maximum, and standard deviation related to the distribution of metals in the plant essential oils, were computed. Other statistical analyses, including correlation analysis (CA), hierarchical cluster analysis (HCA), and non-metric multidimensional scaling (nMDS) were performed using SPSS v.28. The non-metric multidimensional scaling represents an ordination technique where a small number of axes are explicitly reduced using rank orders (distances) for ordination. The advantage of using nMDS ordination is associated with optimization of stress represented by the difference between the distance that, in our case, is lower than 0.01. We applied ANOVA analysis using the Tukey Post Hoc test and a one-sample T-test for variance and graphic visualization using Origin Pro 2022. The analytical data were presented as mean ± standard deviation for mean values of each plant species (regardless of producer). The network plot was performed using the Person product–moment correlation matrix as a database, respectively Fruchterman–Reingold parametrization and conditional values of health risk assessment indices (*r* > 0.5 for THQ and *r* > 0.2 for CR). The analysis indicated the relationship between toxic, micro, and macronutrients graphically.

## 3. Results

### 3.1. Toxic Metal Content in Plant Essential Oils

The statistical data (ANOVA) of toxic metal content in plant essential oils analysis highlighted no significant differences between concentrations of the investigated metals based on the producer’s criteria. The elemental level of toxic trace elements in plant essential oils is illustrated in Figure 1. The highest elemental As, Cd, Hg, and Pb content were found in the L1, M1, M6, and J2 samples, 0.018, 0.017, 0.612 and 0.133 mg/kg, respectively.

We evaluated each essential oil grouped by plant species in the following order As, Cd, Hg, and Pb. The results showed that for *Thymus vulgaris* L., the extreme values in C5, C2, C1, and C6 (0.011, 0.014, 0.076 and 0.050 mg/kg) with a Hg > Pb > Cd > As decreasing average. The maximum values in *Lavandula augustifolia* Mill. Were found in L1, L2, L1, and L3 (0.018, 0.010, 0.062, and 0.069 mg/kg) and the mean values were decreased as follows: Hg > Pb > As > Cd. For *Mentha* × *pipperita* L., the Hg level was over six times higher than the international recommended limits described in European Pharmacopoeia [35] in M1 (0.612 mg/kg). Except for this sample, the average Hg content varied between 0.049 and 0.065 mg/kg. Below limits were Pb, As, and Cd (M1 = 0.133, M1 = 0.010 and M2 = 0.005 mg/kg). The mean values for *Pinus sylvestris* L. essential oil were arranged as follows Hg > Pb > Cd > As (0.063, 0.047, 0.014 and 0.005 mg/kg) and the maximum content were found in P7, P5, P7, and P5 (0.05, 0.014, 0.063, and 0.047 mg/kg). The maximum toxic metals concentration measured in *Juniperus communis* L. were in J1, J2, J4, and J3 (0.014, 0.017, 0.057, and 0.052), with a mean decreasing order of Hg > Pb > As > Cd.

### 3.2. Microelements Content in Plant Essential Oils

The Al, Cr, Mn, Co, Ni, Cu, and Zn content investigated in our studied samples showed high variability among producers and plant oil types, with results illustrated in Figure 2. The Zn registered a significant level in the case of M6 (32.76 mg/kg); the values were much lower in the other products, C6, L2, J3, and P4 (3.50, 2.14, 0.84, and 0.74 mg/kg). We noted that Al had the maximum value in L2, C3, M4, P2, and J2 (10.09, 9.11, 7.75, 5.13, and 3.79 mg/kg). Although it is not essential for humans, Ni has been studied for potentially harmful effects on health [36], and in our case the extreme mean values were found in C4, M2, L4, P5, and I3 (1.44, 1.00, 0.94, 0.67, and 0.13 mg/kg). The maximum Cr level was measured in C4, M4, L4, P5, and J2 (2.55, 1.94, 1.83, 1.65, and 0.84 mg/kg). For Cu, the highest value was in J6 (1.43 mg/kg), and the other samples’ concentrations decreased as follows C1, M3, P3, and L6 (0.37, 0.15, 0.12 and 0.11 mg/kg). Low mean Mn and Co levels were found in analyzed samples, considering the highest values in M4 (0.25 mg/kg) and L4 (0.04 mg/kg).

### 3.3. Macroelements Content in Plant Essential Oils

Figure 3 shows the Na and the Mg concentration in the investigated plant essential oil samples, ranging (in mg/kg) between 0.110 (M2) and 4.281 (M4) and 0.002 (P1) and 3.373 (L5). The Mg varied between producers in the case of *Thymus vulgaris* L. in the range of 0.177 mg/kg (C2) and 1.068 mg/kg (C6), respectively. Na had a minimum value in C2 (0.372 mg/kg), and the maximum level was found in C5 (2.444 mg/kg). The *Lavandula augustifolia Mill*. emphasized extreme Mg (3.373 mg/kg) in L5 and Na (3.885 mg/kg) in L1 essential oil samples with a mean value calculated between producers of 0.839 ± 1.042 mg/kg (Mg) and 1.805 ± 1.092 mg/kg (Na). The *Mentha* × *pipperita* L. samples showed the highest Mg and Na (1.733 and 4.281 mg/kg) in the M4 samples, with a mean value per macroelement of 0.570 ± 0.573 and 1.438 ± 1.295 mg/kg. For *Pinus sylvestris* L. essential oil, the P5 contained the largest levels of Mg and Na (3.289 and 2.951 mg/kg), and the lowest macroelements level were found in P1 samples with a mean of 0.853 ± 1.046 mg/kg (Mg) and 1.470 ± 0.806 mg/kg (Na). Finally, *Juniperuss communis* L. samples showed a maximum amount of Mg and Na of 1.193 and 2.307 mg/kg in J5 and J3, and mean values were 0.566 ± 0.325, 1.419 ± 0.625 mg/kg.

### 3.4. Multivariate Analysis

An ANOVA analysis was applied to the thirty-four different oil types and producer brands. The quantified levels of heavy metals resulted in no significant differences at *p* < 0.05 between them. The results of the correlation analysis considering all samples are illustrated in Figure 4a, demonstrating a solid statistically significant relationship (*p* < 0.001, two-tailed test of significance) between Cr versus Ni, Co, Mn, respectively, and Zn versus Pb.

Analyzing each essential oil based on plant species, we noted a significant positive relationship between Mn, Co, Ni, and Cr for *Thymus vulgaris* L., *Lavandula augustifolia* Mill., *Mentha* × *pipperita* L., and *Pinus sylvestris* L (Figure 4b–e). For *Pinus sylvestris* L., there was a robust positive relationship between Pb, Mg, and Na versus Cr, Mn, Co, and Ni (Figure 4e). For *Juniperus communis* L., we noted a not very strong but significant relationship, both positive and negative (Figure 4g).

Based on the maximum concentration of heavy metals for each category of plant oil products, the hierarchical cluster analysis divided the metals and the plant species into four clusters (Figure 5a,b). *Thymus vulgaris* L. and *Lavandula augustifolia* Mill. were associated in the first cluster, with *Pinus sylvestris* L. and *Juniperus communis* L. in the second group. The *Mentha* × *pipperita* L. differentiated significantly from the other two groups. The heavy metal groups emphasized the Al followed by Zn with the highest maximum metal concentration in different brands of plant essential oils. The macroelements were associated with microelements Cr and Ni in one distinct group. Investigations based on the producer highlighted three groups differentiating two producers (Solaris and Herbalsana) from others (Figure 5c).

A statistical model based on non-metric multidimensional scaling analysis (nMDS) was used to verify plant species–specific bioaccumulation patterns. The model showed that Adams, Doterra, Hypericum, and Steaua Divina essential oils are extracted from plants that accumulate high micro and macroelements (Cu, Mn, Mg, Na) levels. The Solaris plant extract is mainly associated with low Ni and Zn microelements amounts. The low amounts of As, Pb, Cd, and Co appear to be associated in one dimension and characterized the Fares products. The last loading indicated a possible association between Hg and Herbalsana essential oil products (Figure 6).

### 3.5. Health Risk Assessment

The chronic daily intake (CDI) for Al and Cr was significantly higher than the rest of the elements, especially in *Lavandula augustifolia* Mill. (0.00158–0.00285 µg/kg bw/day) and *Mentha* × *pipperita* L. (0.00072–0.00571 µg/kg bw/day) oil (Table 2). The lowest CDI values were registered in the case of As (1.97 × 10^−7^–2.05 × 10^−5^ µg/kg bw/day) and Cd (8.31 × 10^−8^–1.50 × 10^−5^ µg/kg bw/day). The results showed higher values for THQ_Cr (3.87 × 10^−5^–6.11 × 10^−4^) and THQ_Hg (1.77 × 10^−5^–1.36 × 10^−4^) compared to the rest of the metals (Figure 7c). In the end, the cancer risk values for Ni were the highest (2.02 × 10^−9^–7.89 × 10^−7^) (Figure 7d).

The network plot shows the involvement of each heavy metal inclination in inducing risk for human consumers. Three groups were calculated for THQ, including As, Hg, and Cd in the first group. In the second were Cr, Mn, Ni, Co, and in the third were associated Zn and Al. The correlation matrices indicated significant THQ values between Al–Zn (*r* = 0.64), Cr–Mn (*r* = 0.67), Cr–Co (*r* = 0.72), Cr–Ni (*r* = 0.63), Cr–Hg (*r* = 0.58), Mn–Co (*r* = 0.97), Mn–Ni (*r* = 0.95), Co–Ni (*r* = 0.96), As–Hg and Cd–Hg (*r* = 0.63) (Figure 7a). For CR, we observed a single cluster including toxic metals Hg, Cr, and As. The significant correlation coefficients were calculated only between Cd–Hg and As–Hg (0.63) (Figure 7b). The results indicated possible cumulative effects of toxic metals that increase the risk for human consumers.

## 4. Discussions

Some essential minerals are indispensable for healthy human nutrition; their deficit and even high abnormal levels can induce functional disruption [37]. Due to their importance in metabolic functions, the growth and the formation of bones, and the nervous system’s function, Ca, Mg, Na, Fe, and Mn are considered essential for human health [37,38]. International law does not present the safety limits for toxic metals in an expressway in medicinal/culinary plants. We found several recommendations imposed by European Pharmacopoeia (1, 0.1 and 5 mg/kg for Cd, Hg, and Pb) [35], FAO/WHO (1, 0.30 and 10 mg/kg for As, Cd, and Pb) [39], and WHO (0.2 and 0.3 mg/kg for Cd and Pb) [40].

The sample M1 (*Mentha* × *pipperita* L.) with its plant origin in India could be used internally as a supplement, demonstrating very high mercury contamination (Figure 1). In this country, more than eight species of *Mentha* are cultivated for pharmaceutical purposes [41]. Unfortunately, India is facing high mercury levels in agricultural soils, which can be assimilated differently in plants [42]. The origins of Hg in agricultural soils are related to polluted freshwater from rivers used for irrigation and fertilizers that are the pathway for toxic metals in aromatic plants [43]. The wild mint accumulated seasonally heavy metals in the underground organs. Previous results indicated that Hg is associated with sulfate-reducing bacteria in the environment, resulting in the highly toxic methylmercury, which can be easily transferred through plants to the food chain [44]. In plants, Hg stress affects metabolic processes and growth regardless of the stages of development [45]. The plants have multiple methods of fighting heavy metal toxicity, such as osmotic regulation, chelation, or compartmentalization. Thus, plant organs can assimilate high amounts of toxic metals, further transferred to derived products, as is in our case, the essential oils. 

In the last years, various studies indicated severe pollution in multiple environments caused by intensive industrialization, mining, agricultural practices, and poor management of city waste in Romania [46], which increased the susceptibility of foodstuff affected by pollution [47]. Even if lead did not exceed the recommended limit, it had high concentrations in all essential oils, especially in *Mentha* × *pipperita* L., reaching a maximum in the M6 sample with plant origins in Romania (Table 1). Lead has been reported to accumulate in plant roots and less in aerial organs. Still, the deposits are consistent at high concentrations, even in aerial plant tissues, due to disruption of the plasma membranes. A recent study on plant growth for essential oil content indicated negligible or no transfer of Pb from plant to water during the distillation process [48]. The authors discussed a competition between Cd, Pb, and Cu, but we observed different behavior between toxic metals in our samples, either synergic or antagonistic. Other studies found a nonlinear relationship between the elemental concentration from plants and the essential oil extracted in the case of lead and copper [49]. The possibility that essential oils from plants cultivated on highly contaminated soils contain very high Pb, Cu, and Cd was also discussed [50]. Additionally, we noted that the concentration of metals varied in essential oils with varying origins of plants and species. For all the analyzed oil samples, the chronic daily intake value was significantly lower than the provisional tolerable daily dose recommended by the World Health Organization [51] and the European Food Safety Authority [52]. 

Environmental conditions are responsible for elemental content in plants [53], including geo-climatic localization, soil type, and plant development stage. Those influences regulate plant composition and essential oil yields, such as controlling the plant’s internal physiology to synthesize different chemicals specific to the same species [48]. The plant’s essential oil composition can vary greatly depending on the habitat, location, climatic conditions, and soil biology. Some authors suggested that aluminum can be conditioned by contamination sources, such as metallic containers for storage during harvesting and handling processes [54]. This metal contamination is usually associated with chromium from similar sources. The hydrodistillation method for the extraction of essential oils could be the main reason for lower levels of metals in the composition of plant essential oils [55]. However, even if these are obtained by distillation, metals can enter the oils during storage and production processes (e.g., steel containers). As a fact, 10–15% of the population in industrial countries are affected by metal hypersensitivity, mainly nickel, cobalt, and chromium, and they are subjected to various diseases. Up to 17% of women and 3% of men are allergic to Ni, and 1–3% of people are allergic to Co and Cr [56]. In the present study, we chose these elements since they are well-known allergenic metals besides other possible emergent allergens like Al, Cu, Mn, and Zn. We also noted differences between elemental profiles of plant essential oils based on the lowest level of microelements (e.g., Cu) in *Pinus sylvestris* L. and *Juniperus communis* L. and the highest in *Mentha* × *pipperita* L (Figure 2). For the post-harvesting of the plants during the distillation and the extraction of essential oils, cooper containers were used, which can explain the occurrence of the low Cu concentration. This explanation can interpret the lack of a significant correlation between Cu and other elements (Figure 4). 

Chromium is an essential mineral with beneficial effects in regulating insulin action and improving carbohydrate and lipid metabolism [57]. According to our measurements, the highest mean concentration level was in *Thymus vulgaris* L. and *Mentha* × *pipperita* L., explained by possible contamination during the technological process. The higher chromium content can also suggest the influence of the geochemical feature of the region where the plant was primarily collected rather than the human-induced polluted environment. The non-carcinogenic health risk associated with exposure to heavy metals was assessed by calculating the target hazard quotients (THQs) and the hazard index (HI) using the method proposed by the US-EPA [58]. By summing the THQs, the HI was calculated. Because the used method allows quantifying total chromium without performing speciation of this element, it was used in the worst-case scenario for THQ_Cr, by assuming that the entire Cr content was CrIV. The RfD_CrIV (3 µg/kg bw/day) was used instead of RfD_CrIII (1500 µg/kg bw/day), which led to higher values for THQ_Cr. Our analysis indicated the highest indices in *Pinus sylvestris* L., *Mentha* × *pipperita* L., and *Lavandula augustifolia* Mill. The results support this statement for all investigated essential oil based on the robust relationship between Cr and Ni (Figure 4). Nickel is widely distributed in the environment in natural forms, but its provenance through soil application of low-quality fertilizers and micronutrients is also discussed [59]. The significant differences for all metals investigated assessed based on the health risk dataset for the same type of essential oil were found depending on the manufacturer. For all the analyzed oil samples, the hazard index values were considerably lower than 1, which indicated that all values fell within the safety range [60]. Thus, the content of the analyzed elements in the essential oils does not pose any significant risk to consumers’ health. 

The network plot calculated using the cancer risk index showed that nickel was associated with toxic metals. The potential cancer risk based on lifetime exposure to carcinogens (Ni, As, Cd, Hg, and Pb) does not appear to modify the subject’s risk of cancer (CR) [60], even if, for Ni, the values are very high. The results are a consequence of the higher nickel content registered in these oils and the higher dose recommended by the producers for internal consumption. The higher values were obtained in one of the *Lavandula augustifolia* Mill., samples and two *Thymus vulgaris* L. essential oil samples, varying on producers. In the case of THQ, the CR values for arsenic are very different depending on the manufacturers. The cumulative cancer risk was obtained by summing the CR values for each metal. For all the analyzed oils, the value of this parameter was <10^−6^, which indicates that the cancer risk associated with metal exposure through oil ingestion can be ignored [60]. Thus, the analyzed oils can be safely consumed in the doses recommended by the manufacturers, and the content of heavy metals does not pose a significant risk to the consumer’s health.

Zinc is a vital micronutrient required for the structural and functional integrity of the biological membrane and the detoxification of free radicals, with an essential role in maintaining cell and organ integrity and as an antioxidant and anti-inflammatory agent [61]. According to the World Health Organization, zinc deficiency is a worldwide public health problem that affects 31%, with the prevalence rates ranging from 4 to 73% in various regions of the world population [62]. Here, we can discuss the soil-related origins of metal concentration transferred from plants to essential oil. The M6 sample with sources in Romania had a maximum concentration of more than 3.5 times more than other maximum M3 and plants with similar origins. The *Mentha* × *pipperita* L. experiment indicated that the plants could stabilize the metals at the root level for a short period, showing tolerance even to toxic metals [26]. The plants were affected by long term exposure but showed an efficient mechanism of excluding the metals by roots only in the case of As and Pb, for which the transfer to aerial plants was insignificant.

Multivariate analyses, including Pearson product–moment correlation, hierarchical cluster analysis, and non-metric multidimensional scaling, examined the heavy metals contamination of natural and anthropogenic sources in plant essential oils. The statistical analyses emphasized a strong relationship between macronutrients and toxic metals, which was expected for the essential oils with worldwide origins. Even so, only for *Pinus sylvestris* L. was there a significant association between micro and macronutrients and their concentration in EOs comparable to other studies [63]. The role of macroelements in plant physiology was frequently discussed, even in those used for essential oil extraction [63]. Mg deficiency-induced foliar necrosis is responsible for old leaves’ decline and dieback since the manganese in the plant is a mobile microelement that can be transferred from old leaves to younger ones, limiting the effects of deficiency. The toxicity of Mn is not often clearly identified; still, it can vary between species and even between different locations. 

## 5. Final Consideration and Future Perspective

Diverse literature studied the extraction process, composition, and beneficial proprieties of plants collected from accessible sites around the world from which were extracted the EOs. We used various search engines to cover as much literature as possible, and we did not find significant interest in investigating differences among producers based on elemental profiles. Thus, in the present investigation, we tested the essential oils accessible on the market as the final product with worldwide origins and various producers. Since we did not extract the oil from plants, and we analyzed plant EOs that were already bottled, this can be discussed as a limitation of the present study. The advantages of the research can be associated with improving the current knowledge regarding the total amount of elemental metal levels ingested in one day from different sources or, more practically, choosing the right product for each need based on macro and micronutrient content. Consumers can even evaluate the products based on toxic profiles, or the producers can even improve extraction techniques to diminish or to reduce this content. 

Innovative tools and natural, eco-friendly solutions based on raw materials are continuously included in the pharmaceutical and in the food industry to adjust to consumers’ needs. The popularization of these chemical molecules with antioxidant and antimicrobial activity promotes their usage on a large scale. Still, concerns are raised by conventional agriculture, where environmental pollution exceeds tolerable limits in many ecosystems. The properties of the final product, as in our case, plant essential oils, presented to the consumer need to be carefully analyzed for toxic elemental content. In the present study, we observed a significant correlation between micronutrient content and toxic elemental level that can be interpreted through prevalent bounds between elements or similar intake pathways of metals in plants. All mentions indicate that the analyzed metals are in concentrations that do not threaten consumers’ health. They can be safely used for internal consumption by following the dosage recommended by their producers, even if we found abnormal toxic metal concentrations in some samples.

Several metals are associated with the technological process, causing contamination of the final products, and this is a reason why the producers need to invest in advanced technology. More than metal pollution, we are interested in analyzing an extensive range of contaminants linked with agricultural practices, such as pesticides, hydrocarbons, persistent organics, or antibiotics. The evaluation of EOs toxicity indicated by statistical modelling through in vivo cytotoxicity studies represents a key factor that should be explored in future research. Further, we considered, evaluated, and compared the antibacterial, antioxidant, and cytotoxic activities of various essential oil brands based on differences in elemental composition. Our research regarding the comparative study of different essential oil brands can help producers understand the limitation and the advantages of their products. It can also be used as a reference for acquiring the raw material used in the extraction of EOs, considering the peculiarities of plant species for the bioabsorption of metals. 

## Figures and Tables

**Figure 1 nutrients-14-02363-f001:**
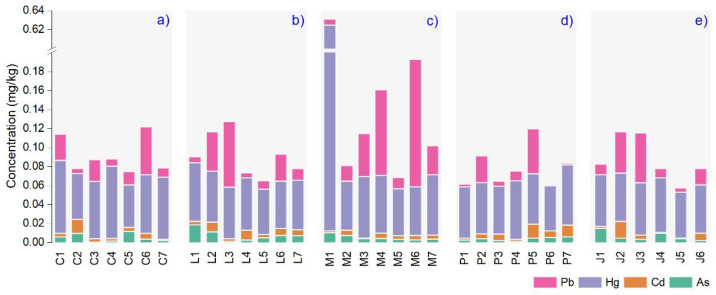
The toxic elemental content (mg/kg) quantified in each essential oil: (**a**) *Thymus vulgaris* L., (**b**) *Lavandula augustifolia* Mill., (**c**) *Mentha* × *pipperita* L., (**d**) *Pinus sylvestris* L., (**e**) *Juniperus communis* L.

**Figure 2 nutrients-14-02363-f002:**
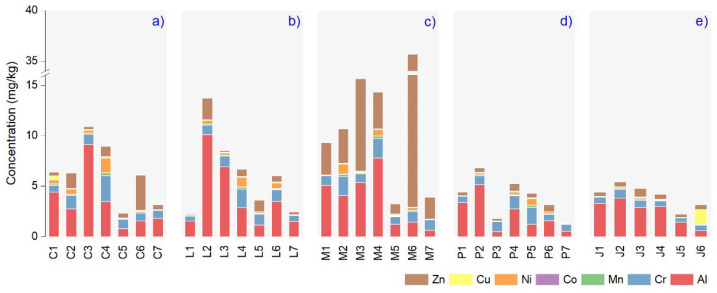
The micronutrients elemental levels (mg/kg) in samples analyzed: (**a**) *Thymus vulgaris* L., (**b**) *Lavandula augustifolia* Mill., (**c**) *Mentha* × *pipperita* L., (**d**) *Pinus sylvestris* L., (**e**) *Juniperus communis* L.

**Figure 3 nutrients-14-02363-f003:**
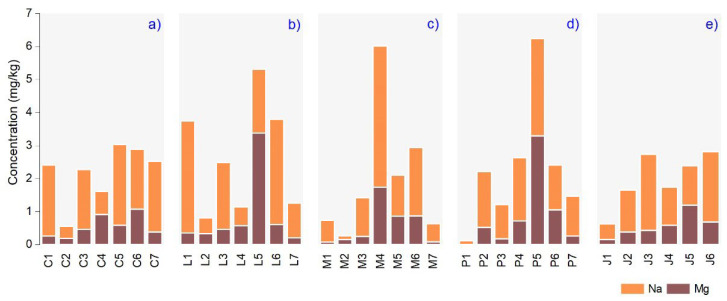
The macronutrients elemental levels (mg/kg) measured in the samples analyzed: (**a**) *Thymus vulgaris* L., (**b**) *Lavandula augustifolia* Mill., (**c**) *Mentha* × *pipperita* L., (**d**) *Pinus sylvestris* L., (**e**) *Juniperus communis* L.

**Figure 4 nutrients-14-02363-f004:**
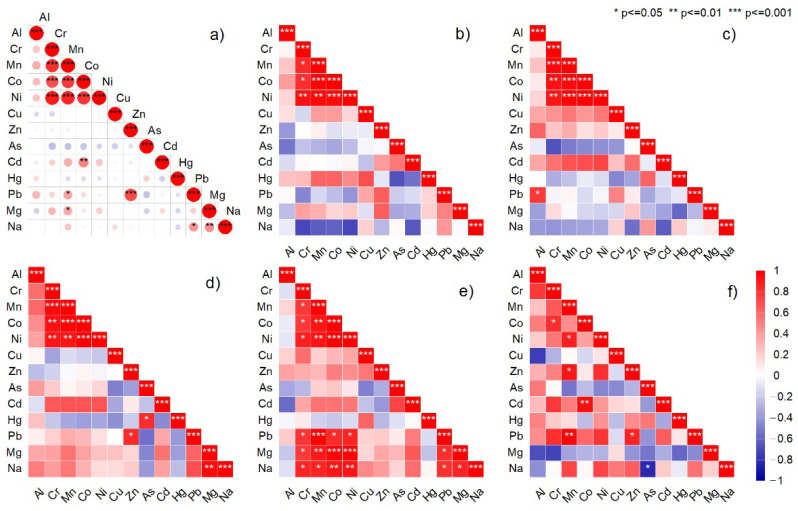
Correlation coefficient matrix of heavy metals concentration (mg/kg) in plant essential oil products; (**a**) All essential oil; (**b**) *Thymus vulgaris* L., (**c**) *Lavandula augustifolia* Mill., (**d**) *Mentha* × *pipperita* L., (**e**) *Pinus sylvestris* L., and (**f**) *Juniperus communis* L.

**Figure 5 nutrients-14-02363-f005:**
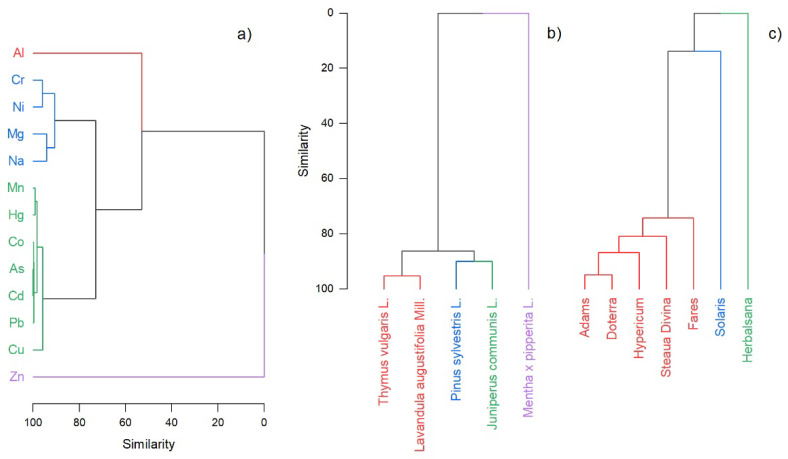
Hierarchical cluster analysis of heavy metals in different plant essential oil product categories: (**a**) based on the maximum heavy metal concentration, (**b**) association based on plant species used for essential oil extraction, and (**c**) grouping based on the producer.

**Figure 6 nutrients-14-02363-f006:**
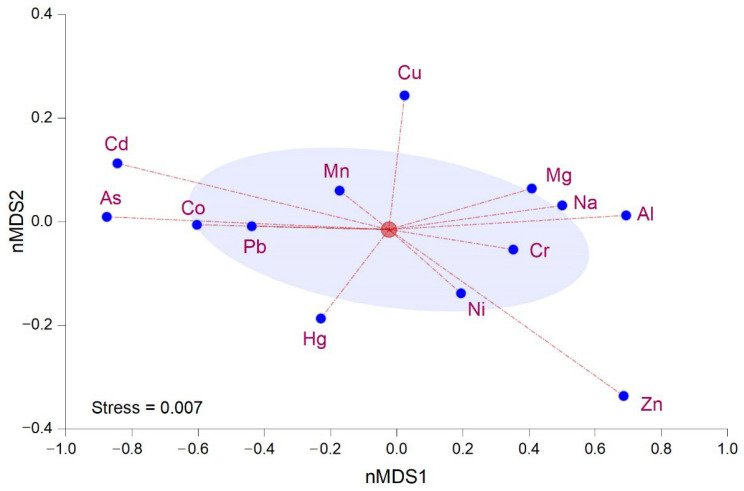
Non-metric dimensional analysis of heavy metals in plant essential oil products.

**Figure 7 nutrients-14-02363-f007:**
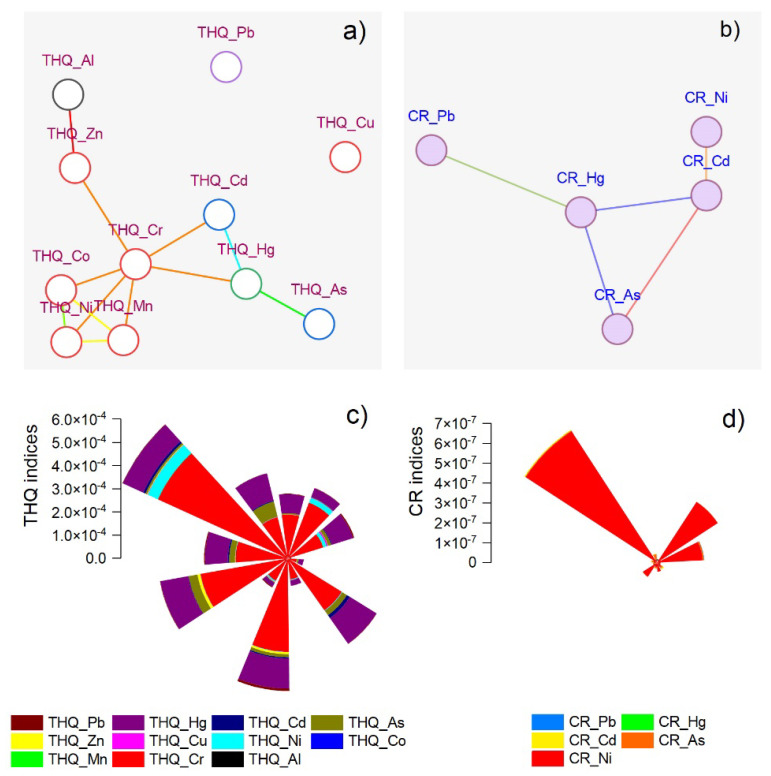
The network plot related to THQ index for 11 heavy metals and 34 plant oil products (**a**); Hazard Index (HI) resulted by summing the target hazard quotient (THQ) for each metal presented in panel (**c**); The network plot related to CR index for 11 heavy metals and 34 plant oil products (**b**); Total cancer risk calculated by summing the cancer risk for each metal presented in panel (**d**).

**Table 1 nutrients-14-02363-t001:** Selected samples with linked attributes.

Species	COD	Producer	Origin	∑metals	Species	COD	Producer	Origin	∑metals
*Thymus vulgaris* L.	C1	Adams	India	8.9075	*Lavandula augustifolia* Mill.	L1	Adams	India	6.0237
*Satureja hortensis* L.	C2	Hypericum	Romania	6.9294	*Lavandula augustifolia* Mill.	L2	Hypericum	Romania	14.6322
*Satureja hortensis* L.	C3	Steaua Divina	Romania	13.2498	*Lavandula augustifolia* Mill.	L3	Steaua Divina	Romania	11.1439
*Thymus vulgaris* L.	C4	Doterra	Salt Lake City, UT, USA	10.6565	*Lavandula augustifolia* Mill.	L4	Doterra	Salt Lake City, UT, USA	7.8597
*Thymus vulgaris* L.	C5	Herbalsana	UE	5.4272	*Lavandula augustifolia* Mill.	L5	Herbalsana	UE	8.9778
*Thymus vulgaris* L.	C6	Solaris	Romania	9.0870	*Lavandula hybrida*	L6	Solaris	Romania	9.9007
*Thymus vulgaris* L.	C7	Fares	Romania	5.7259	*Lavandula augustifolia* Mill.	L7	Fares	NaN	3.7830
*Mentha × pipperita* L.	M1	Adams	India	10.6569	*Pinus sylvestris* L.	P1	Adams	Austria	4.5618
*Mentha × pipperita* L.	M2	Hypericum	Romania	10.9964	*Pinus sylvestris* L.	P2	Hypericum	Romania	9.1167
*Mentha × pipperita* L.	M3	Steaua Divina	Romania	17.1625	*Pinus sylvestris* L.	P3	Steaua Divina	NaN	3.0284
*Mentha × pipperita* L.	M4	Doterra	Salt Lake City, UT, USA	20.4874	*Pinus sylvestris* L.	P4	Doterra	Salt Lake City, UT, USA	7.9497
*Mentha × pipperita* L.	M5	Herbalsana	UE	5.4209	*Pinus sylvestris* L.	P5	Herbalsana	UE	10.6651
*Mentha × pipperita* L.	M6	Solaris	Romania	38.8149	*Pinus sylvestris* L.	P6	Solaris	NaN	5.5946
*Mentha × pipperita* L.	M7	Fares	Romania	4.6063	*Pinus sylvestris* L.	P7	Fares	Romania	2.8702
*Juniperus communis* L.	J1	Adams	Croatia	5.0788					
*Juniperus communis* L.	J2	Hypericum	Romania	7.1587					
*Juniperus communis* L.	J3	Steaua Divina	Romania	7.6018					
*Juniperus communis* L.	J4	Doterra	Salt Lake City, UT, USA	5.9878					
*Juniperus communis* L.	J5	Herbalsana	UE	4.6808					
*Juniperus communis* L.	J6	Solaris	NaN	6.0233					

**Table 2 nutrients-14-02363-t002:** The chronic daily intake (CDI) of trace and major elements associated with the essential oil’s ingestion.

CDI—Chronic Daily Intake (µg/kg bw/day)											
Type	Al	Cr	Mn	Co	Ni	Cu	Zn	As	Cd	Hg	Pb
Thyme	0.00313	0.00047	9.18 × 10^−5^	1.55 × 10^−5^	27.3958 × 10^−5^	0.000268	0.000291	4.28 × 10^−6^	2.41 × 10^−6^	5.46 × 10^−5^	1.97 × 10^−5^
*Thymus vulgaris*	0.00105	0.00077	7.07 × 10^−5^	1.17 × 10^−5^	43.3903 × 10^−5^	3.85 × 10^−5^	0.000322	4.34 × 10^−7^	7.91 × 10^−7^	2.29 × 10^−5^	2.21 × 10^−6^
*Thymus vulgaris*	0.00128	0.00056	1.86 × 10^−5^	2.29 × 10^−6^	2.4002 × 10^−6^	2.99 × 10^−5^	0.000336	1.35 × 10^−6^	7.33 × 10^−7^	4.66 × 10^−5^	6.9 × 10^−6^
*Lavandula augustifolia*	0.00171	0.00055	1.94 × 10^−5^	3.53 × 10^−6^	1.42882 × 10^−5^	1.29 × 10^−5^	0.000132	2.05 × 10^−5^	3.72 × 10^−6^	6.88 × 10^−5^	6.76 × 10^−6^
*Lavandula augustifolia*	0.00285	0.00183	0.000178	4.64 × 10^−5^	93.8768 × 10^−5^	5.25 × 10^−5^	0.000727	2.34 × 10^−6^	1.05 × 10^−5^	5.49 × 10^−5^	5.42 × 10^−6^
*Lavandula augustifolia*	0.00158	0.00067	1.61 × 10^−5^	3.38 × 10^−6^	1.97719 × 10^−5^	3.03 × 10^−5^	0.000309	7.45 × 10^−6^	6.62 × 10^−6^	5.61 × 10^−5^	1.31 × 10^−5^
*Mentha* × *pipperita*	0.00571	0.00113	3.72 × 10^−5^	4.43 × 10^−6^	2.61226 × 10^−6^	2.11 × 10^−5^	0.003654	1.16 × 10^−5^	2.23 × 10^−6^	6.95 × 10^−5^	7.28 × 10^−6^
*Mentha* × *pipperita*	0.00129	0.00032	4.17 × 10^−5^	3.63 × 10^−6^	10.0599 × 10^−5^	1.15 × 10^−5^	0.000609	6.75 × 10^−7^	9.54 × 10^−7^	1.01 × 10^−5^	1.5 × 10^−5^
*Mentha* × *pipperita*	0.00072	0.00121	2.54 × 10^−5^	4.55 × 10^−6^	2.75431 × 10^−5^	1.66 × 10^−5^	0.002512	4.04 × 10^−6^	4.77 × 10^−6^	7.41 × 10^−5^	3.53 × 10^−5^
*Pinus sylvestris*	0.00057	0.00027	1.3 × 10^−5^	2.09 × 10^−6^	5.38326 × 10^−5^	2.56 × 10^−5^	0.000155	1.97 × 10^−7^	4.21 × 10^−7^	1.28 × 10^−5^	2.14 × 10^−6^
*Pinus sylvestris*	0.00063	0.00082	3.58 × 10^−5^	5.91 × 10^−6^	2.92926 × 10^−5^	3.4 × 10^−5^	5.51 × 10^−5^	7.07 × 10^−6^	1.5 × 10^−5^	7.74 × 10^−5^	1.95 × 10^−6^
*Juniperus communis*	0.00062	0.00012	9.52 × 10^−6^	1.17 × 10^−6^	1.79451 × 10^−5^	8.47 × 10^−6^	9.72 × 10^−5^	2.05 × 10^−6^	8.31 × 10^−8^	1.2 × 10^−5^	2.07 × 10^−6^
PTDI (µg/kg bw/day) *	142.86	300.00	66.66	-	2.80	50–500	300–1000	2.14	0.83	0.57	3.57

* PTDI—provisional tolerable daily intake, according to World Health Organization (WHO) and European Food Safety Authority (EFSA) recommendations.

## Data Availability

All relevant data to the study are included in the article.

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
