# Peer review of "Assessing the Health Risk and the Metal Content of Thirty-Four Plant Essential Oils Using the ICP-MS Technique"

_nutrients, 2022, doi:10.3390/nu14122363_

Round 1

Reviewer 1 Report

Nutrients (Manuscript ID: nutrients-1735174), Comments to the Authors:

Title: Assessing the health risk and chemical profile of thirty-four plant essential oils using the ICP-MS technique

Comments

The submitted manuscript investigated the essential oil samples, compared similar products between seven producers, and the results indicated a wide variation of metal content. The recommended limits imposed by European Union regulations for medicinal plants were exceeded only in Mentha x pipperita (Adams, 0.61 mg/kg). Except for Thymus vulgaris, the multivariate analysis showed a strong correlation between toxic and microelements. The authors verified plant species-specific bioaccumulation patterns with non-metric multidimensional scaling analysis. The results show a model where most essential oils are associated with a high microelements content. According to target hazard quotients and carcinogenic risk, the health risk assessment indicated a safely use for pharmaceutical and edible functions in doses recommended by the manufacturers.

The authors should respond to the following comments:

  1. The title is confusing, the authors did not assess the phytochemical component of the oil they assesses the metal content. The title should be changed
  2. The abstract is disorganized and should be rephrased to provide clear picture of the manuscript results and show the value of the work. The authors should include numerical data to show the importance of their work.
  3. The authors should clearly indicate the application of their findings for the oil industry and how their findings can improve the quality and safety of the marketed oils.
  4. The authors should add few statements on the future perspectives of their work.
  5. The authors should compare their work with previous research to show the value of the submitted manuscript.
  6. The introduction is lengthy and should be rephrased.
  7. The discussing is foul of unnecessary information and should be rephrased to represent a clear cohesive idea on the importance of the obtained results.  

Author Response

Dear anonymous reviewer,

We address sincere gratitude for the careful and thorough reading of the manuscript and the valuable comments and constructive suggestions that helped improve the draft's quality. We have revised our manuscript according to the recommendations, and its final version is enclosed. Point-by-point responses to the comments are listed below. We hope that the revised manuscript is acceptable for publishing.

Reviewer 1

The authors should respond to the following comments:

1. The title is confusing, the authors did not assess the phytochemical component of the oil they assesses the metal content. The title should be changed

Response: We changed the title to "Assessing the health risk and chemical profile of thirty-four plant essential oils using the ICP-MS technique". Thank you!

2. The abstract is disorganized and should be rephrased to provide clear picture of the manuscript results and show the value of the work. The authors should include numerical data to show the importance of their work.

Response: The abstract was rewritten and we included numerical data. Thank you for the recommendation.

3. The authors should clearly indicate the application of their findings for the oil industry and how their findings can improve the quality and safety of the marketed oils.

Response: We added a statement indicating the application of our findings to the oil industry and how our results can be used to increase the quality and safety of the essential oils market in the final chapter, "Final consideration and future perspective"

4. The authors should add few statements on the future perspectives of their work.

Response: The chapter "Final consideration and future perspective" included several statements regarding future research interests. Thank you for the suggestion.

5. The authors should compare their work with previous research to show the value of the submitted manuscript.

Response: We searched the scientific literature for similar studies comparing various essential oils based on the producer. We found no relevant result that includes a similar product as we used. Thus, we compared our results to articles describing essential oils metal content from regions closer to those indicated on bottles used in the present investigation.

6. The introduction is lengthy and should be rephrased.

Response: We eliminated paragraphs from the introductions, and the section was rephrased. Thank you for the suggestion.

7. The discussing is foul of unnecessary information and should be rephrased to represent a clear cohesive idea on the importance of the obtained results.

Response: We eliminated unnecessary information from the Discussion chapter. Thank you!

Sincerely yours,

Constantin NECHITA

Reviewer 2 Report

The manuscript "Assessing the health risk and chemical profile of thirty-four plant essential oils using the ICP-MS technique" provides relevant information on the risk of consuming plants with traces of toxic elements.

The manuscript is well structured, and the conclusions are clear. My suggestion is to put a section on future perspectives, where it is indicated that to verify the toxicity that was indicated by means of chemical tests and computational models, preliminary in vitro cytotoxicity studies should be carried out.

Author Response

Dear anonymous reviewer,

We address sincere gratitude for the careful and thorough reading of the manuscript and the valuable comments and constructive suggestions that helped improve the draft's quality. We have revised our manuscript according to the recommendations, and its final version is enclosed. Point-by-point responses to the comments are listed below. We hope that the revised manuscript is acceptable for publishing.

Reviewer_2

The manuscript "Assessing the health risk and chemical profile of thirty-four plant essential oils using the ICP-MS technique" provides relevant information on the risk of consuming plants with traces of toxic elements.

Response: Thank you very much for evaluating our manuscript.

The manuscript is well structured, and the conclusions are clear. My suggestion is to put a section on future perspectives, where it is indicated that to verify the toxicity that was indicated by means of chemical tests and computational models, preliminary in vitro cytotoxicity studies should be carried out.

Response: We added to the manuscript a final chapter entitled "Final consideration and future perspective". Here we enclosed the main result, which consists in observing a significant relationship between micronutrient content and toxic elemental level and future research interest. Thank you for the suggestion which improved our work.

Sincerely yours,

Constantin NECHITA

Round 2

Reviewer 1 Report

Nutrients (Manuscript ID: nutrients-1735174), Comments to the Authors:

Title: Assessing the health risk and chemical profile of thirty-four plant essential oils using the ICP-MS technique

Comments

After reading the authors' response to my comments, I believe the authors responded to all my remarks and the manuscript can be accepted for publication. 

This manuscript is a resubmission of an earlier submission. The following is a list of the peer review reports and author responses from that submission.